# Advanced Respiratory Failure Requiring Tracheostomy—A Marker of Unfavourable Prognosis after Heart Transplantation

**DOI:** 10.3390/diagnostics14080851

**Published:** 2024-04-20

**Authors:** Marta Załęska-Kocięcka, Marco Morosin, Jonathan Dutton, Rita Fernandez Garda, Katarzyna Piotrowska, Nicholas Lees, Tuan-Chen Aw, Diana Garcia Saez, Ana Hurtado Doce

**Affiliations:** 1Department of Mechanical Circulatory Support and Transplantation, National Institute of Cardiology, 04-628 Warsaw, Poland; 2Department of Anaesthesia and Critical Care, Harefield Hospital, Royal Brompton and Harefield NHS Foundation Trust, London SW3 6NP, UK; jonathn.dutton@gmail.com (J.D.); rita_ferga@hotmail.com (R.F.G.); n.lees@rbht.nhs.uk (N.L.); t.aw@rbht.nhs.uk (T.-C.A.); a.hurtadodoce@rbht.nhs.uk (A.H.D.); 3Department of Anaesthesia and Critical Care, Royal Brompton Hospital, Royal Brompton and Harefield NHS Foundation Trust, London SW3 6PY, UK; marco.morosin01@gmail.com; 4Department of Quantitative Methods and Information Technology, Kozminsky University, 03-301 Warsaw, Poland; kpiotrowska@kozminski.edu.pl; 5Department of Cardiothoracic Transplantation and Mechanical Circulatory Support, Harefield Hospital, Royal Brompton and Harefield NHS Foundation Trust, London SW3 6PY, UK; d.garciasaez@rbht.nhs.uk

**Keywords:** heart transplantation, tracheostomy, ex vivo perfusion

## Abstract

Advanced respiratory failure with tracheostomy requirement is common in heart recipients. The aim of the study is to assess the tracheostomy rate after orthotopic heart transplantation and identify the subgroups of patients with the highest need for tracheostomy and these groups’ association with mortality at a single centre through a retrospective analysis of 140 consecutive patients transplanted between December 2012 and July 2018. As many as 28.6% heart recipients suffered from advanced respiratory failure with a need for tracheostomy that was performed after a median time of 11.5 days post-transplant. Tracheostomy was associated with a history of stroke (OR 3.4; 95% CI) 1.32–8.86; *p* = 0.012), previous sternotomy (OR 2.5; 95% CI 1.18–5.32; *p* = 0.017), longer cardiopulmonary bypass time (OR 1.01; 95% CI 1.00–1.01; *p* = 0.007) as well as primary graft failure (OR 6.79; 95% CI2.93–15.71; *p* < 0.001), need of renal replacement therapy (OR 19.2; 95% 2.53–146; *p* = 0.004) and daily mean SOFA score up to 72 h (OR 1.50; 95% 1.23–1.71; *p* < 0.01). One-year mortality was significantly higher in patients requiring a tracheostomy vs. those not requiring one during their hospital stay (50% vs. 16%, *p* < 0.001). The need for tracheostomy in heart transplant recipients was 30% in our study. Advanced respiratory failure was associated with over 3-fold greater 1-year mortality. Thus, tracheostomy placement may be regarded as a marker of unfavourable prognosis.

## 1. Introduction

Advanced respiratory failure is a common complication following cardiothoracic surgery associated with increased mortality, diminished quality of life, and great economic burden. Tracheostomy (TT) is a common procedure performed in patients requiring prolonged mechanical ventilation (MV) due to a severe respiratory insufficiency after cardiac surgery. Despite being an invasive procedure that creates a surgical airway in the cervical trachea, the risk profile in either general cardiothoracic or transplant population is low and commonly limited to non-severe complications such as mild bleeding [1,2]. However, severe adverse events such as major haemorrhage or pneumothorax cannot be neglected [1,2]. At the same time, tracheostomy brings many potential benefits such as a reduction in sedation requirement, the avoidance of laryngeal injury, airway protection, improvement in patient comfort, and the allowance of gradual ventilatory weaning and physical rehabilitation.

The need for tracheostomy occurs in up to 3.3% of cardiac surgical patients and is associated with unfavourable outcomes [1,2]. Recently, Wang et al. identified the following risk factors for the need of tracheostomy in general cardiac surgical patients: mixed valve surgery and coronary artery bypass grafting (CABG), aortic surgery, renal insufficiency, diabetes mellitus, chronic obstructive pulmonary disease (COPD), pulmonary edema, age > 60 years, emergent surgery, and previous stroke [3]. Data regarding adult heart transplant population are limited to the general thoracic population with an overrepresentation of lung transplant patients [4]. The aim of the study is to assess tracheostomy rate after orthotopic heart transplantation and identify the subgroups of patients with the highest need for tracheostomy and these groups’ association with mortality.

## 2. Methods

### 2.1. Study

This is a retrospective study registered within the Trust Research Office at the Royal Brompton and Harefield NHS Foundation trust. The study was conducted in compliance with the Declaration of Helsinki. Due to the retrospective nature of the study, patients’ informed consent was waived. Data regarding hyperlactatemia in the studied population were published previously [5].

### 2.2. Population

One hundred fifty-three consecutive patients receiving cardiac transplants between December 2012 and July 2018 in a single tertiary centre were analysed. Patients were included in the study if they received a single-organ heart transplantation, with donor organs preserved in a state of ex vivo perfusion during retrieval, using the TransMedics Organ Care System (TransMedics Inc., Boston, MA, USA). Patients who were younger than 16 years old, died during the procedure, or had previous tracheostomy or tracheostomy performed in the operating room were also excluded. Based on the given criteria, as many as 140 patients were included in the final analysis according to patients flow chart (Figure 1). Donor matching was performed in accordance with the NHSBT Heart Transplantation: Organ Allocation policy (POL228/10) [6].

### 2.3. Tracheostomy

The decision for tracheostomy was based on clinician judgement according to the patient’s best interest and guided by a multidisciplinary team in view of difficult or expected prolonged respiratory wean. The default selection choice was percutaneous dilatational tracheostomy (PDT).

### 2.4. Data Collection

Retrospective data were collected by Clinical Informatics from major clinical systems including the following: Philips IntelliSpace Critical Care & Anaesthesia system, patient administration, laboratory, electronic drug prescription and internal National Institute for Cardiovascular Outcomes Research (NICOR) dataset.

### 2.5. Statistical Analysis

The frequency distributions of categorical variables are presented in percentages. And for the purpose of the analysis, in case of the categorical variables, dummy coding was applied. Continuous data are represented as mean ± standard deviation [SD] or median with interquartile range [IQR] depending on data distribution. Shapiro–Wilk test was used to assess for normality. For the 2 independent groups, the means of normally distributed data are compared using the Student’s *t*-test; the mean ranks of non-normally distributed data are compared using the Mann–Whitney *U*-test. Chi square test was used to test the relationships between categorical variables. Univariate and multivariate logistic regression was used to test the statistical significance of the association between the selected variables and the need for tracheostomy with *p* < 0.05 as well as to estimate the odds ratio (OR). Kaplan–Meier curves were utilised to present survival probability. The curves were compared using the log–rank test, a *p*-value < 0.05 was considered statistically significant, and all reported *p*-values were 2-sided. The Cox proportional hazard model was applied to analyse the effects of the need for tracheostomy as well as other risk factors on the overall survival time. The statistical analysis was performed using IBM^®^SPSS Statistics 28.

## 3. Results

In total, 153 consecutive single-organ heart transplant recipients were reviewed between December 2012 and July 2018. Given the inclusion and exclusion criteria, final analysis included the remaining 140 heart transplant patients as displayed in the flow chart (Figure 1).

The average age of the recipients was 45 years (16–72) and 31.4% (*n* = 44) were females. The leading causes of end-stage heart failure were dilated cardiomyopathy (*n* = 74, 52.9%) and ischaemic cardiomyopathy (*n* = 40, 28.6%). The majority (81.4%) were transplanted whilst being listed as urgent (UHAS), with 6.4% classified as super-urgent (SUHAS) according to the Cardiothoracic Advisory Group adjudication panel; the remainder were transplanted as non-urgent (NUHAS) (POL229/6) [7]. At the time of the heart transplant, 42.1% (*n* = 59) were on mechanical circulatory support (MCS). Mean follow-up time in the study was 980 mean SD 747 days.

As many as 65% (*n* = 91) of the heart transplant patients required prolonged mechanical ventilation (>48 h) and nearly one third had a tracheostomy performed (TT group; *n* = 40; 28.6%) after a median time of 11.5 days from ITU admission. The majority of TT were performed via percutaneous route (90%; *n* = 36). Patients requiring TT had a significantly higher rate of stroke/TIA history (*p* = 0.009), previous sternotomy (*p* = 0.008), and longer cold ischaemia time (*p* = 0.045) and cardiopulmonary bypass time (*p* = 0.009). Comparison of the baseline, perioperative and early postoperative characteristics between the TT and non-TT groups are displayed in Table 1.

The following baseline and perioperative risk factors for TT were identified as significantly associated with the need of tracheostomy: history of stroke/TIA (OR 3.4; 95% CI 1.32–8.86; *p* = 0.012), previous sternotomy (OR 2.5; 95% 1.18–5.32; *p* = 0.017), and longer cardiopulmonary bypass (CBP) time (OR 1.01; 95% CI 1–1.01; *p* = 0.007) (Table 2).

Postoperative outcome analysis revealed that the patients suffering from issues other than respiratory organ system failure also had a greater need of having TT performed. Nearly all patients (98%) in the TT subgroup required renal replacement therapy (RRT) after heart transplantation compared to 67% of patients in the non-TT group (*p* < 0.001, Table 1). Furthermore, the rate of primary graft failure requiring VA ECMO support was nearly four times higher when compared to the non-TT group (*p* < 0.001, Table 1). The lactate, bilirubin and alanine transaminotransferase levels were also significantly higher in the first 72 h post transplantation (Table 1) in the TT group. Differences in SOFA score up to 72 h from ITU admission reflected that the TT group developed overall greater extent of end-organ dysfunction compared to non-TT (all *p* < 0.02 Table 1).

In an unadjusted analysis of renal failure requiring RRT, PGF requiring VA ECMO support and mean SOFA score in the first 72 h from ITU admission after heart transplantation were associated with the need of TT in the later course (Table 2).

After adjustment for preoperative and clinical variables, PGF requiring VA ECMO support as well mean SOFA score in the first 72 h of ITU stay were significantly related to the risk of needing TT (Table 3).

As expected, the TT patients experienced longer mechanical ventilation (*p* < 0.001) and consequently longer ITU stay (28.5 (IQ 21–49.8) vs. 6 (IQ 4–8) days in non-TT group *p* < 0.001) (Table 1). There were no life-threatening complications related to the procedure: major bleeding requiring intervention, pneumothorax, pneumomediastinum, tube misplacement, or cardiac arrest.

Despite a comparable 30-day mortality between the TT and non-TT group, 90-day and 1-year mortality were over 3-fold higher in those requiring TT (Table 1). The Kaplan–Meyer curve showed significantly lower survival over the study period in patients requiring TT (Figure 2, *p* < 0.001). However, after adjustment for other factors, Cox regression analysis indicated that only PGF was related with mortality but not TT (Table 4).

Furthermore, results regarding the time of TT performance were inconclusive. Time from ITU admission to TT insertion was shorter in 90-day survivors compared to non-survivors (10 (IQ 7.8–13.3) days in survivors vs. 13.5 (IQ 10.5–18.3); *p* = 0.030), but there was no difference between 1-year survivors and non-survivors (10 (8–13.8) vs. 12.5 (IQ 9.0–17.5); *p* = 0.125). The results of the Kaplan–Meyer analysis showed comparable outcomes between early (<14) days and late (≥14 days) TT recipients (Figure 3). Yet, early and steep curves divergence in the chart should not be neglected, especially given the worse short-term (90 day) prognosis in late strategy (Figure 3).

## 4. Discussion

The present study reported predictors and implications of advanced respiratory failure with TT requirement after cardiac transplantation. The issue was evaluated and investigated in a broad spectrum of cardiothoracic population. However, to our knowledge, the present study is the first dedicated to heart transplant recipients. It was found that as many as one third of our population suffer from significant respiratory insufficiency with the need to perform TT. The findings support the study published by Pilarczyk et al., who reported that 29.8% of thoracic transplant patients required TT [4]. At such rate, the need for TT is about 10-fold higher compared to general cardiac surgery [2]. Thus, heart transplant candidates should be informed about the risk involved.

Discrepancies in TT prevalence between general cardiac surgery population and OHT reflect diverse clinical profiles of the patients and subsequently, risk factors associated with TT. To our knowledge, this is the first study to identify tracheostomy risk factors in the single heart transplant population.

In terms of preoperative risk factors associated with the need of TT, the following were found: previous cardiac surgery and stroke/TIA. In our study, past medical history of previous sternotomy in OHT population was common given the fact that as many as 35% were bridged to transplant with a durable LVAD. Despite there being no reports indicating that re-sternotomy is linked to TT after cardiac surgery, an abundance of evidence proves its association with prolonged mechanical ventilation [7]. Following the fact that the incidence of respiratory failure after undergoing cardiac surgery is frequent and often long lasting, this factor should be taken into consideration in heart transplant recipients [8,9].

The current study confirmed that a history of previous stroke is a well-established risk factor for the need of TT after cardiac surgery, and this also applies to the heart transplant population [10]. In contrast to general cardiac surgery patients, baseline renal insufficiency, diabetes mellitus, chronic obstructive pulmonary disease (COPD), and age >60 years were not found to be associated with a higher incidence of TT in our transplant population [10]. However, these conditions are considered relative if not absolute contraindications for heart transplantation. So, their prevalence in our study was expected to be low. For instance, COPD, a well-established risk factor for both prolonged mechanical ventilation and TT, was reported in only 4.3% patients (Table 1). Such rate is in line with previous reports regarding heart transplant recipients, yet it is significantly lower compared to the cardiac surgery population where it varies between 8% after CABG and 20% after valvular heart surgery [11].

Another potential risk factor is that the intraoperative amount of red blood cell (RBC) transfusion was also not confirmed in the present study, while it was identified as independent risk factors for TT in a population of patients undergoing acute type A aortic dissection surgery and lung transplant [12,13].

Unfortunately, other potential risk factors were missed, for instance, phrenic nerve palsy. This complication is common in lung transplant recipients and is associated with prolonged mechanical ventilation and increased morbidity [12]. Interestingly, it is a potentially curable condition with diaphragm pacing [14].

Postoperatively, the need for TT was associated with ongoing or worsening multiorgan failure. Apart from obvious respiratory failure, all of our patients developed at least one additional organ failure such us kidney failure requiring RRT or circulatory failure requiring ECMO support. Liver enzymes were also significantly elevated in the TT recipients. Thus, markers of physiological derangement such as lactate level and SOFA score significantly outstood in the TT subgroup.

Consequently, the need of TT was associated with markedly worse prognosis. Unfavourable outcomes in heart transplant patients requiring TT are consistent with existing studies in various populations. One-year mortality at the level of 50% in heart transplant recipients with TT is close to the range reported in either cardiac surgery (37–42%) or general ITU population (36–46%) [10,15,16,17]. Given the burden of accompanying multiorgan syndrome, in terms of cause-and-effect relationship, tracheostomy should be regarded as a reflection of an advanced underlying condition rather than a causative factor. And this was supported by the Cox regression analysis, which, once adjusted for other risk factors, did not reveal TT as significantly related to outcomes (Table 4). Similar observations were made by Krebs et al. in their extensive analysis covering 14,600 general cardiac surgery patients, out of whom 309 required TT [10]. When analysing the impact of tracheostomy on long-term mortality while controlling for other risk factors, tracheostomy itself did not predict increased mortality either.

In general, the total one-year mortality was high, however, given the population characteristics, it was in line with the United Kingdom (UK) data. Over 80% transplantation were in urgent mode, and 42% recipients had preOHT mechanical circulatory support with over 30% having cf-LVAD (Table 1). According to UK national report from 1 April 2014 to 31 March 2018, one-year survival rate after heart transplant was 82.4% [18]. However, rates varied depending on the presence and type of mechanical circulatory support. The lowest rate was shown in patients supported with long term LVAD 67.3% (56.4–76.0), followed by short term LVAD 73.7% (62.0–82.4) [18]. Survival rate of unsupported patients was as high as 87.2% (83.6–90.2) [18]. A large analysis by Whitbread JJ et al. including 20,113 transplant recipients with 45% supported with LVAD showed that among those who died after transplantation, patients with LVADs on average died sooner (1.8 years) than patients without LVADs (3.0 years; *p* < 0.01) [19]. On multivariable analysis, patients with LVADs had a 44% higher mortality risk within the first 3 months post-transplant (*p* = 0.03) [19]. And the trend was persistent through the next 9 months as patients with LVADs had statistically borderline 21% increased risk of death within the first year post-transplant conditional on 3 months of post-transplant survival (*p* = 0.06) [19].

Despite that, understanding the high risk of having TT procedure performed and its translation into poor long-term prognosis provide important guidance for patients undergoing the procedure. Studies show that having tracheostomy affects not only speech and communication but also wellbeing, quality of life, and body image; and is related to stigma and social withdrawal [20]. In the light of available data, both heart transplant patients and their families should be informed at the moment of giving an informed consent.

Furthermore, the procedure itself was free of severe adverse events. However, the study did not aim to assess either short- or long-term complications related to tracheostomy.

Available data evaluating the impact of the timing of TT insertion are inconclusive. The results of the randomized TracMan trial showed that early TT in general ICU population was not associated with improvement either in the primary outcome defined as 30-day mortality or in secondary outcomes including the length of stay in the critical care [21]. Yet many studies, albeit nonrandomized, put into question such findings. In the recent retrospective study covering over 900 cardiac surgery patients, it was concluded that early TT (<7 days) may provide better clinical outcomes, with lower mortality and morbidity rates [22]. A research group led by A. Vuylsteke reported longer mechanical ventilation time and higher complication rates with delayed strategy in cardiothoracic ICU patients [23]. In the present study, the Kaplan–Meyer analysis did not show significant differences in survival between early vs. late approach. However, a notable divergence of survival lines should not be neglected given the better 90-day survival in the early approach. Furthermore, given the fact that the need for TT appears to be remarkably higher after OHT compared to general ITU population, it will be sensible to use a pre-emptive strategy to maximize the advantages of tracheostomy. However, given the data paucity on the subject, this requires further investigation.

## 5. Conclusions

This study shows that severe respiratory failure with tracheostomy requirement should be regarded as a marker of severely complicated postoperative condition with remarkably limited prognosis. Although non-significant, data suggest possible advantages of early vs. late approach. Despite being a low-risk procedure, its high rate and clinical impact urge further investigation for risk factors. The information about the risk of having tracheostomy performed and its association with poor survival after heart transplantation should be part of an informed consent.

### Limitations

This is a single centre retrospective study, and thus can only ever be hypothesis-generating and may not be generalisable. However, this is also the first study analysing the need and risk factors for tracheostomy dedicated exclusively to heart transplant recipients. There are still many aspects requiring further investigation such as other relevant risk factors. In that case, phrenic nerve palsy should be of interest. Unfortunately, neither the current study nor any other, except for case studies, evaluated that factor in heart transplant recipients. Yet, the complication is common after lung transplantation (up to 40%) with remarkable impact on prolonged ventilation, risk of reintubation, and need for non-invasive procedures [10]. Furthermore, little is known in terms of mid- and long-term complication rates in organ recipients. Increased rate of sternal wound infections after TT in immunosuppressed transplant patients call for an evaluation.

## Figures and Tables

**Figure 1 diagnostics-14-00851-f001:**
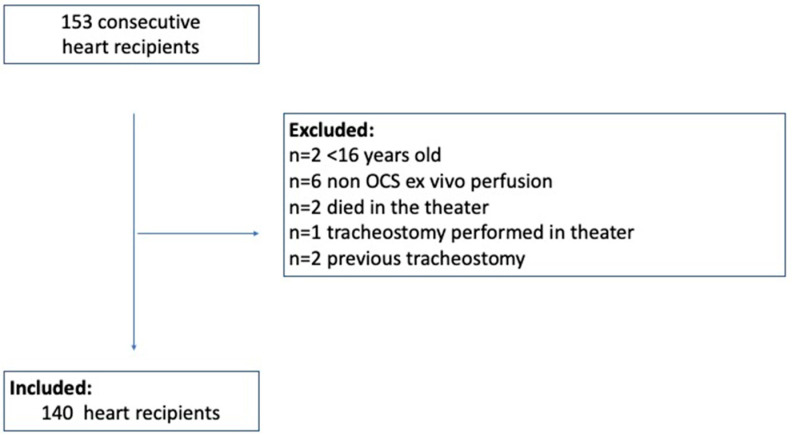
Flowchart of patients included in the study.

**Figure 2 diagnostics-14-00851-f002:**
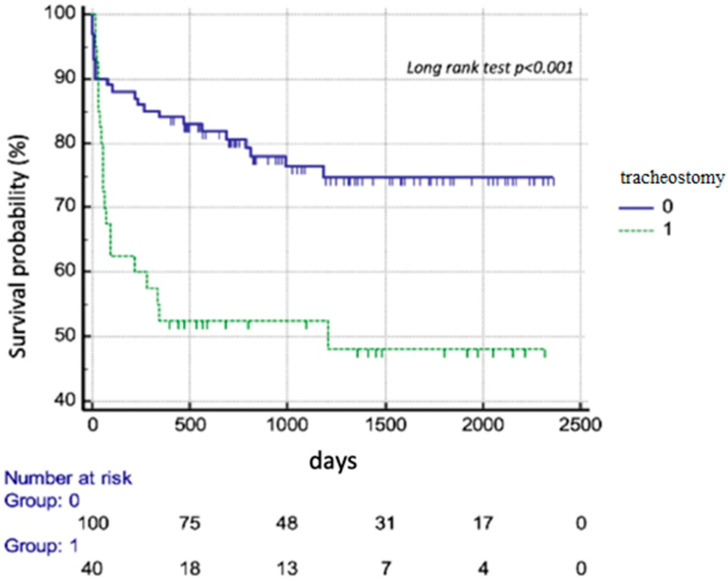
Survival probability of heart transplant patients with and without tracheostomy.

**Figure 3 diagnostics-14-00851-f003:**
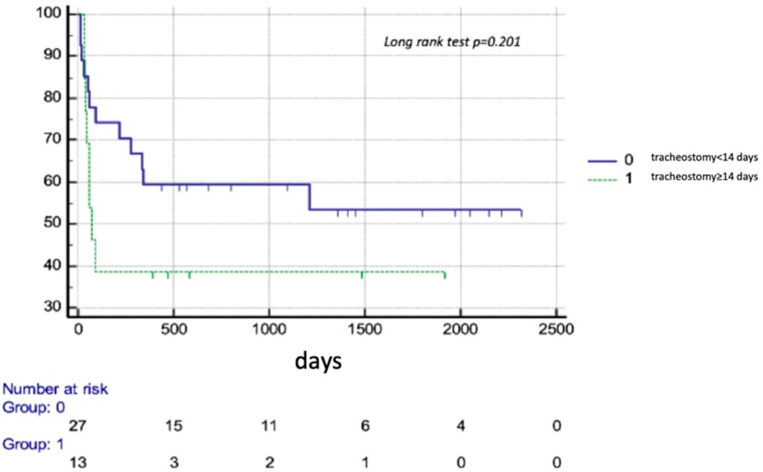
Survival probability of early (<14 days) vs. late (≥14 days) trachesotomy recipients among heart transplant patients.

**Table 1 diagnostics-14-00851-t001:** Baseline, perioperative and outcome characteristics between TT and non-TT patients.

Variables	Non-TT (*n* = 100)	TT (*n* = 40)	*p*-Values
*Preoperative risk factors*			
Age, years (SD)	44.9 (13.5)	48.0 (14.4)	0.235
Sex (female)	32%	30%	0.819
Hypertension	14%	13%	0.817
History of stroke	10%	28%	0.009
Coronary artery disease	24%	30%	0.463
Diabetes	9%	5%	0.427
Smoking	19%	8%	0.091
Previous sternotomy	33%	58%	0.008
PreOHT MCS	38%	53%	0.117
COPD	5%	2.5%	0.509
Creatinine, μmol/L	95.39 (31.9)	88.2 (27.12)	0.212
Pulmonary resistance (woods unit; IQ)	2.2 (1.4, 3.1)	2.19 (1.5, 2.9)	0.739
Long term LVAD	35%	34%	0.913
*Transplantation-related risk factors*			
OHT urgency:			
Elective	15%	5%	0.102
Urgent	80%	85%	0.492
Emergent	5%	10%	0.276
Donor age, years	40.8 (12.0)	39.5 (11.4)	0.574
Recipient/Donor BSA ratio	0.96 (0.90,1.02)	0.99 (0.91,1.05)	0.063
Organ Care System duration, min (SD)	259.5 (81.8)	270.6 (71.9)	0.456
Cold ischaemia time, min (SD)	78.8 (13.8)	84.5 (13.9)	0.045
Cardiopulmonary bypass time, min (IQ)	161.0 (142.5, 197.8)	183.5 (152.5, 255.5)	0.009
*Postoperative risk factors*			
Mean RBC transfusion up 72 h, mL (IQ)	846.0 (612.7, 2185.0)	1096.67 (784.7, 1706.0)	0.795
Maximum lactate in 24 h	10.3 (3.8)	11.9 (5.0)	0.033
Maximum lactate in 48 h	3.26 (2.37)	3.66 (2.21)	0.361
Maximum lactate in 72 h	2.08 (1.14)	2.29 (1.15)	0.331
Maximum bilirubin in 24 h	36.00 (25.3, 48.0)	40.00 (25.3, 70.0)	0.206
Maximum bilirubin in 48 h	22.0 (14.0, 39.0)	36.00 (18.0, 55.0)	0.011
Maximum bilirubin in 72 h	17.0 (11.0, 27.0)	28.00 (18.0, 44.3)	0.001
Maximum ALT in 24 h, IU (IQ)	42.0 (28.0, 70.3)	72.00 (37.0, 208.8)	0.001
Maximum ALT in 48 h, IU (IQ)	42.0 (31.0, 72.0)	128.00 (41.0, 211.0)	<0.001
Maximum ALT in 72 h, IU (IQ)	41.0 (27.0, 66.0)	108.00 (35.3, 237.8)	<0.001
Inotropic score 24 h (IQ)	10.0 (7.0, 15.0)	15.00 (10.0, 20.0)	<0.001
Inotropic score 48 h (IQ)	8.0 (5.0, 12.0)	12.00 (10.0, 18.0)	<0.001
Inotropic score 72 h (IQ)	5.0 (2.0, 10.0)	11.00 (8.5, 16.0)	<0.001
Maximum SOFA score in 24 h (SD)	15.3 (3.1)	16.8 (2.4)	0.007
Maximum SOFA score in 48 h (SD)	13.1 (4.5)	16.5 (2.4)	<0.001
Maximum SOFA score in 72 h (SD)	11.0 (5.5)	16.0 (2.7)	<0.001
*Outcomes*			
Primary graft failure	14%	52.5%	<0.001
RRT	67%	98%	<0.001
Duration of RRT, days (IQ)	4.5 (1.5, 10)	30.2 (20.2, 68.6)	<0.001
Duration of mechanical ventilation, days (IQ)	2.07 (1.4, 4.0)	22.29 (14.1, 55.2)	<0.001
Duration of ITU stay, days (IQ)	6.00 (4.0, 8.0)	28.50 (21.0, 49.8)	<0.001
30-day mortality	10%	13%	0.666
90-day mortality	11%	35%	<0.001
1-year mortality	16%	50%	<0.001

ALT alanine transferase; BSA body surface area; COPD chronic obstructive pulmonary disease; ITU intensive therapy unit; IQ interquartile; IU international units; LVAD left ventricle assist device; MCS mechanical circulatory support; OHT orthotopic heart transplantation; RBC red blood cell; RRT renal replacement therapy; SD standard deviation; SOFA sequential organ failure assessment; TT tracheostomy.

**Table 2 diagnostics-14-00851-t002:** Univariate logistic regression of tracheostomy risk factors.

Variables	OR	95% CI	*p*-Value
History of stroke/TIA	3.41	1.32–8.86	0.012
Pre-transplant sternotomy	2.5	1.18–5.32	0.017
Cardiopulmonary bypass time	1.01	1–1.01	0.007
Mean SOFA up to 72 h	1.50	1.23–1.71	<0.01
RRT in ITU	19.2	2.53–146	0.004
Primary graft failure	6.79	2.93–15.71	<0.001

ITU intensive therapy unit; RRT renal replacement therapy; SOFA sequential organ failure assessment; TIA transient ischaemic attack.

**Table 3 diagnostics-14-00851-t003:** Multivariate logistic regression for tracheostomy risk factors.

Variables	OR	95% CI	*p*-Value
Age	0.99	0.96–1.03	0.758
Female sex	0.80	0.26–2.41	0.689
History of stroke/TIA	2.63	0.73–9.43	0.138
Pre-transplant sternotomy	3.14	1.15–8.56	0.025
Cardiopulmonary bypass time	0.99	0.99–1.00	0.169
Mean SOFA up to 72 h	1.28	1.04–1.57	0.018
RRT in ITU	7.92	0.86–73.42	0.068
Primary graft failure	4.5	1.37–214.77	0.013

ITU intensive therapy unit; RRT renal replacement therapy; SOFA sequential organ failure assessment; TIA transient ischaemic attack.

**Table 4 diagnostics-14-00851-t004:** Multivariate Cox regression analysis of factors contributing to overall survival.

Variables	HR	95% CI	*p*-Value
Tracheostomy	0.70	0.31–1.60	0.396
Pre-transplant sternotomy	1.72	0.85–3.50	0.133
Cardiopulmonary bypass time	1.0	0.99–1.0	0.656
Mean SOFA up to 72 h	1.05	0.90–1.21	0.546
RRT in ITU	1.31	0.38–4.53	0.667
Primary graft failure	6.96	2.92–16.56	<0.001
History of stroke	2.1	0.97–4.43	0.058

ITU intensive therapy unit; RRT renal replacement therapy; SOFA sequential organ failure assessment; TIA transient ischaemic attack.

## Data Availability

Data is contained within the article.

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
