# Peer review of "Advanced Respiratory Failure Requiring Tracheostomy—A Marker of Unfavourable Prognosis after Heart Transplantation"

_diagnostics, 2024, doi:10.3390/diagnostics14080851_

Round 1

Reviewer 1 Report

Comments and Suggestions for Authors

I am grateful to the editor for the opportunity to review the manuscript by ZaÅ‚Ä™ska-KociÄ™cka et al “Advanced respiratory failure requiring tracheostomy - a marker of unfavorable prognosis after heart transplantation”. In this manuscript, the authors study a rather narrow problem - predictors of the need for tracheostomy in patients after heart transplantation, as well as the possible impact on the prognosis of this procedure. The result was generally expected - among the predictors of the need for tracheostomy were factors associated with both the initial comorbidity and the severity of the surgical intervention, as well as the development of postoperative complications. It is quite expected that among complicated patients on prolonged mechanical ventilation and requiring tracheostomy, the annual prognosis is worse. The authors showed this in a separate cohort of patients after heart transplantation, which is a new scientific fact.

While reviewing, I had the following comments:

1. The goal in the Abstract is poorly formulated and consists of two separate sentences. I propose to adjust it similarly to the purpose of the study in the Introduction section.

2. It is necessary to more clearly formulate the novelty of the research. The authors first state that "Data regarding heart transplant population are limited and suggest markedly higher tracheostomy utilization, reaching up to 30%" (lines 56-58). And then they set the goal “The aim of the study is to describe tracheostomy rate after orthotopic heart transplantation” (lines 58-59). Why set such a goal if the tracheostomy rate has already been studied in this category of patients?

3. The authors refer to a flowchart, but it is not included in the manuscript.

4. In the general cohort of patients, the annual mortality rate after heart transplantation is at least 25% (more patients died in the theater). Is this percentage too high? Is it possible to compare this data with other centers?

5. The clinical significance of this study remains unclear. It is clear that tracheostomy is performed during long-term mechanical ventilation due to the complicated course of the postoperative period. However, it does not have an independent effect on the prognosis, so is there a need for such an analysis as carried out in this article.

6. Apparently, the article showed the safety of tracheostomy in this category of patients, since there were no complications associated with it that worsened the prognosis of patients. I don’t quite understand the separate instructions about the need to inform patients and relatives about the possibility of a trachestomy in case of a complicated course of the postoperative period after heart transplantation. According to the authors’ logic, then this should be done separately for all possible postoperative complications? Or did I understand something wrong?

Comments on the Quality of English Language

No comments

Author Response

I am grateful to the editor for the opportunity to review the manuscript by ZaÅ‚Ä™ska-KociÄ™cka et al “Advanced respiratory failure requiring tracheostomy - a marker of unfavorable prognosis after heart transplantation”. In this manuscript, the authors study a rather narrow problem - predictors of the need for tracheostomy in patients after heart transplantation, as well as the possible impact on the prognosis of this procedure. The result was generally expected - among the predictors of the need for tracheostomy were factors associated with both the initial comorbidity and the severity of the surgical intervention, as well as the development of postoperative complications. It is quite expected that among complicated patients on prolonged mechanical ventilation and requiring tracheostomy, the annual prognosis is worse. The authors showed this in a separate cohort of patients after heart transplantation, which is a new scientific fact.

Thank you that you acknowledged that, given the population studied, the results present new scientific fact. We agree with your comment that it is expected that advanced respiratory failure requiring tracheostomy is associated with poor prognosis. What we believe is worth attention is the rate of the complication that limits significantly outcomes of the heart transplantation- a gold standard therapy in advanced heart failure. Among many complications limiting heart transplant patients outcomes that one is rarely reported.

While reviewing, I had the following comments:

  1. The goal in the Abstract is poorly formulated and consists of two separate sentences. I propose to adjust it similarly to the purpose of the study in the Introduction section.

We appreciate that comment. Changes were incorporated according to your suggestions.
That part of the abstract was changed as following:

ABSTRACT (line 25-27): The aim of the study is to describe tracheostomy rate after orthotopic heart transplantation and identify subgroups of patients at a highest risk of a need for tracheostomy and its association with mortality

  1. It is necessary to more clearly formulate the novelty of the research. The authors first state that "Data regarding heart transplant population are limited and suggest markedly higher tracheostomy utilization, reaching up to 30%" (lines 56-58). And then they set the goal “The aim of the study is to describe tracheostomy rate after orthotopic heart transplantation” (lines 58-59). Why set such a goal if the tracheostomy rate has already been studied in this category of patients?

Indeed, such statement is confusing.  In fact, the cited study by Pilarczyk included 93 thoracic transplant patients with 79 subjects (84.9%) had undergone double lung transplant, 11 (11.8%) had undergone heart transplant, 2 (2.2%) had undergone combined heart-lung transplant, and 1 (1.1%) had undergone combined heart-kidney transplant. Thus, this data cannot be extrapolated to heart transplant population. Thank you for noting that. We propose the following change:

INTRODUCTION (line 66-68): Data regarding adult heart transplant population are limited to general thoracic population with overrepresentation of lung transplant patients [4].

We also rearrange the very beginning of the introduction to give more background to the procedure of tracheostomy

INTRODUCTION (line 42-46) Advanced respiratory failure is a common complication following cardiothoracic surgery associated with increased mortality, diminished quality of life and great economic burden. Tracheostomy (TT) is a common procedure performed in patients requiring prolonged mechanical ventilation (MV) due to severe respiratory insufficiency after cardiac surgery.

  1. The authors refer to a flowchart, but it is not included in the manuscript.

Thank you for your vigiliance. We enclose the flowchart in the Methods section as Fig. 1 in the manuscript.

METHODS (line 85-86) Based on the given criteria as many as 140 patients were included in the final analysis according to patients flow chart (Fig.1)

RESULTS (Part of Line 143-145). Given the inclusion and exclusion criteria final analysis included the remaining 140 heart transplant patients as displayed in flow chart (Fig. 1).

  1. In the general cohort of patients, the annual mortality rate after heart transplantation is at least 25% (more patients died in the theater). Is this percentage too high? Is it possible to compare this data with other centers?

We thank the reviewer for the comment. Based on that we put more emphasis on high risk patient profile in discussion which, we believe, brings valuable background to our data Mortality at the level of 25% (only 2 patients died in the theater) refers to extremely high risk population. In reply to concerns data were compared with other studies. The following part was added:

DISCUSSION (line 293-310). In general, the total one-year mortality was high, however, given the population characteristics, it was in line with United Kingdom (UK) data. Over 80% transplantation were in urgent mode, 42% recipients had preOHT mechanical circulatory support with over 30% having cf-LVAD (Table 1). According to UK national report from 1 April 2014 and 31 March 2018, one- year survival rate after heart transplant was 82.4% []. However, rates varied depending on presence and type of mechanical circulatory support. The lowest rate was shown in patients supported with long term LVAD 67.3 % (56.4 - 76.0), followed by short term LVAD 73.7% (62.0 - 82.4). Survival of unsupported patients was as high as 87.2% (83.6 - 90.2). A large analysis by Whitbread JJ et al. including 20,113 transplant recipients with 45% supported with LVAD showed that among those who died after transplantation, patients with LVADs on average died sooner (1.8 years) than patients without LVADs (3.0 years; p < 0.01). On multivariable analysis, patients with LVADs had a 44% higher mortality risk within the first 3 months posttransplant ( p = 0.03). And the trend was persistent through next 9 months as patients with LVADs had statistically borderline 21% increased risk of death within the first year posttransplant conditional on 3 months of posttransplant survival (p = 0.06).

  1. The clinical significance of this study remains unclear. It is clear that tracheostomy is performed during long-term mechanical ventilation due to the complicated course of the postoperative period. However, it does not have an independent effect on the prognosis, so is there a need for such an analysis as carried out in this article.

As the reviewer stated at the very beginning here we report a new scientific fact. The given rate and the clinical meaning in this unique population is novel and require special attention. High rate of respiratory failure should urge further search of risk factor. The current study identified some risk factor but other potentially relevant, for instance phrenic nerve palsy, were missed. We hope the given data will trigger further search.

To emphasize the clinical significance of the findings additional comment was added to discussion and conclusion with new reference:

DISCUSSION (Line 270-273) Unfortunately, other potential risk factor were missed, for instance phrenic nerve palsy. This complication is common in lung transplant recipients and is associated with prolonged mechanical ventilation and increased morbidity [14]. Interestingly, it is potentially a curable condition with diaphragm pacing [15].

AND

CONCLUSION (line 341-342): Its high rate and clinical impact urge further investigation for risk factors.

  1. Apparently, the article showed the safety of tracheostomy in this category of patients, since there were no complications associated with it that worsened the prognosis of patients. I don’t quite understand the separate instructions about the need to inform patients and relatives about the possibility of a trachestomy in case of a complicated course of the postoperative period after heart transplantation. According to the authors’ logic, then this should be done separately for all possible postoperative complications? Or did I understand something wrong?

We appreciate the reviewer concerns. We are convinced that the rate and clinical impact of tracheostomy urge to make it a part of informed consent as other major complications such bleeding or need for reoperation. Invasive nature of treacheostomy, its profound impact on the ability to communicate and on self-esteem affect patient existence.

Indeed, we agree that the fact of safety requires to be emphasized.

Changes were made accordingly:

DISCUSSION(line 314-316) Studies show having tracheostomy affects not only speech and communication but also wellbeing, quality of life, body image, and is related to stigma and social withdrawal [22].

CONCLUSION (line 341-342) Despite, being a low risk profile procedure, its high rate and clinical impact urge further investigation for risk factors.

  1. Hernández-Hernández MA, Sánchez-Moreno L, Orizaola P, Iturbe D, Álvaréz C, Fernández-Rozas S, González-Novoa V, Llorca J, Hernández JL, Fernández-Torre JL, Parra JA. A prospective evaluation of phrenic nerve injury after lung transplantation: Incidence, risk factors, and analysis of the surgical procedure. J Heart Lung Transplant. 2022 Jan;41(1):50-60. doi: 10.1016/j.healun.2021.09.013. Epub 2021 Oct 3. PMID: 34756781.
  2. Testelmans D, Nafteux P, Van Cromphaut S, Vrijsen B, Vos R, De Leyn P, Decaluwé H, Van Raemdonck D, Verleden GM, Buyse B. Feasibility of diaphragm pacing in patients after bilateral lung transplantation. Clin Transplant. 2017 Dec;31(12). doi: 10.1111/ctr.13134. Epub 2017 Oct 26. PMID: 28990225.
  3. Annual Report on Cardiothoracic Organ Transplantation 2018/2019, NHS Blood and
  4. Whitbread JJ et al. , Etchill EW, Giuliano KA, Suarez-Pierre A, Lawton JS, Hsu S, Sharma K, Choi CW, Higgins RSD, Kilic A. Posttransplant Long-Term Outcomes for Patients with Ventricular Assist Devices on the Heart Transplant Waitlist. ASAIO J. 2022 Aug 1;68(8):1054-1062. doi: 10.1097/MAT.0000000000001611. Epub 2021 Nov 3. PMID: 34743139.
  5. Nakarada-Kordic I, Patterson N, Wrapson J, Reay SD. A Systematic Review of Patient and Caregiver Experiences with a Tracheostomy. 2018 Apr;11(2):175-191. doi: 10.1007/s40271-017-0277-1. PMID: 28914429.

Reviewer 2 Report

Comments and Suggestions for Authors

The manuscript investigates the incidence, risk factors, and outcomes of tracheostomy in heart transplant recipients. It presents a retrospective analysis highlighting a significant association between tracheostomy requirement and adverse outcomes, including higher one-year mortality rates. The study suggests that tracheostomy may serve as an indicator of poor prognosis post-heart transplantation. In general, the analysis process was complete and rigorous. However, there are still some points the authors should address to improve the convincingness of the paper.

Major points

1.     Expand the introduction to include a more detailed discussion of existing literature on tracheostomy in heart transplant recipients, highlighting the novelty and necessity of the current study.

2.     Clarify the selection criteria and potential biases in patient selection. Such as a flowchart.

3.     Provide more detailed analysis on the impact of tracheostomy timing on outcomes.

4.     Address the generalizability of the findings given the single-center nature of the study and discuss more about potential confounders.

Minor points

1.     The p values in Tables 2-4 are shown in the third column, please modify the header.

Author Response

The manuscript investigates the incidence, risk factors, and outcomes of tracheostomy in heart transplant recipients. It presents a retrospective analysis highlighting a significant association between tracheostomy requirement and adverse outcomes, including higher one-year mortality rates. The study suggests that tracheostomy may serve as an indicator of poor prognosis post-heart transplantation. In general, the analysis process was complete and rigorous. However, there are still some points the authors should address to improve the convincingness of the paper.

 Thank you for the summary.

Major points

  1. Expand the introduction to include a more detailed discussion of existing literature on tracheostomy in heart transplant recipients, highlighting the novelty and necessity of the current study.

There is no single study dedicated tracheostomy utility in adult heart transplant patients. Given the population studied, the results present new scientific fact. We included most recent studies. The cited research by Pilarczyk included thoracic transplant patients with (84.9%) had undergone double lung transplant, 11.8% had undergone heart transplant, 2.2% had undergone combined heart-lung transplant, and 1.1) had undergone combined heart-kidney transplant. Thus, this data cannot be extrapolated directly to heart transplant population.

We propose the following change

INTRODUCTION (line 66-68): Data regarding adult heart transplant population are limited to general thoracic population with overrepresentation of lung transplant patients [4].

We also rearrange the very beginning of the introduction to give more background to the procedure of tracheostomy

INTRODUCTION (line 42-46) Advanced respiratory failure is a common complication following cardiothoracic surgery associated with increased mortality, diminished quality of life and great economic burden. Tracheostomy (TT) is a common procedure performed in patients requiring prolonged mechanical ventilation (MV) due to severe respiratory insufficiency after cardiac surgery.

  1. Clarify the selection criteria and potential biases in patient selection. Such as a flowchart.

Thank you for the comment. As suggested the following changes and Fig.1. were added:

METHODS, 2.2. Population (line 79-80): One hundred fifty-three consecutive patients receiving cardiac transplants between December 2012 and July 2018 in a single tertiary centre were analysed.

METHODS, 2.2. Population (line 85-86): Based on the given criteria as many as 140 patients were included in the final analysis according to patients flow chart (Fig.1).

The figure is enclosed in the manuscript.

  1. Provide more detailed analysis on the impact of tracheostomy timing on outcomes.

We appreciate the reviewer remark to emphasize potential impact of tracheostomy timing on patients outcomes. Timing of tracheostomy is reaming an unsolved issue. Yet, the present study delivers arguments in the discussion mainly in the context of short-term survival.

Changes is RESULTS and CONCLUSION sections were incorporated :

RESULTS (line 224-226) : Yet, early and steep curves divergence in the chart should not be neglected, especially given worse short-term (90 day) prognosis in late strategy (Fig.3).

CONCLUSION ( line 342-343) Although non-significant, data suggest possible adventage of early vs late approach.

  1. Address the generalizability of the findings given the single-center nature of the study and discuss more about potential confounders

The current study identified some risk factor but other potentially relevant, for instance phrenic nerve palsy, were missed. We hope the given data will trigger further search.

To emphasize the clinical significance of the findings additional comment was added to discussion and conclusion with new reference:

DISCUSSION (Line 272-275) Unfortunately, other potential risk factor were missed, for instance phrenic nerve palsy. This complication is common in lung transplant recipients and is associated with prolonged mechanical ventilation and increased morbidity [14]. Interestingly, it is potentially a curable condition with diaphragm pacing [15].

AND

CONCLUSION (line 343-344): Its high rate and clinical impact urge further investigation for risk factors.

Minor points

  1. The p values in Tables 2-4 are shown in the third column, please modify the header.

Thank you. This was corrected.

Fig. 1 Study flowchart.

Round 2

Reviewer 1 Report

Comments and Suggestions for Authors

The authors did a great job of correcting the manuscript and answered my questions and comments. I have no other comments.

Comments on the Quality of English Language

No comments

Reviewer 2 Report

Comments and Suggestions for Authors

The authors have implemented the suggested comments to a satisfactory extent.